# A Rapid Head Organ Localization System Based on Clinically Realistic Images: A 3D Two Step Progressive Registration Method with CVH Anatomical Knowledge Mapping

**DOI:** 10.3390/bioengineering11090891

**Published:** 2024-09-01

**Authors:** Changjin Sun, Fei Tong, Junjie Luo, Yuting Wang, Mingwen Ou, Yi Wu, Mingguo Qiu, Wenjing Wu, Yan Gong, Zhongwen Luo, Liang Qiao

**Affiliations:** 1Department of Medical Image, College of Biomedical Engineering and Imaging Medicine, Army Medical University, Chongqing 400038, China; scj@tmmu.edu.cn (C.S.);; 2Army Medical Center of PLA, Army Medical University, Chongqing 400010, China; 3Department of Digital Medicine, College of Biomedical Engineering and Imaging Medicine, Army Medical University, Chongqing 400038, China; 4Department of Radiology, Southwest Hospital, Army Medical University, Chongqing 400038, China

**Keywords:** inter-subject registration, 3D reconstruction, tomographic medical image with clinical quality, organ localization, lightweight processing

## Abstract

Rapid localization of ROI (Region of Interest) for tomographic medical images (TMIs) is an important foundation for efficient image reading, computer-aided education, and well-informed rights of patients. However, due to the multimodality of clinical TMIs, the complexity of anatomy, and the deformation of organs caused by diseases, it is difficult to have a universal and low-cost method for ROI organ localization. This article focuses on actual concerns of TMIs from medical students, engineers, interdisciplinary researchers, and patients, exploring a universal registration method between the clinical CT/MRI dataset and CVH (Chinese Visible Human) to locate the organ ROI in a low-cost and lightweight way. The proposed method is called Two-step Progressive Registration (TSPR), where the first registration adopts “eye–nose triangle” features to determine the spatial orientation, and the second registration adopts the circular contour to determine the spatial scale, ultimately achieving CVH anatomical knowledge automated mapping. Through experimentation with representative clinical TMIs, the registration results are capable of labeling the ROI in the images well and can adapt to the deformation problem of ROI, as well as local extremum problems that are prone to occur in inter-subject registration. Unlike the ideal requirements for TMIs’ data quality in laboratory research, TSPR has good adaptability to incomplete and non-thin-layer quality in real clinical data in a low-cost and lightweight way. This helps medical students, engineers, and interdisciplinary researchers independently browse images, receive computer-aided education, and provide patients with better access to well-informed services, highlighting the potential of digital public health and medical education.

## 1. Introduction

Clinical TMIs such as CT and MRI play an important role in the diagnosis and treatment of diseases. However, due to the abstract nature of the imaging modality and the complexity of the human anatomy, clinical medical images are too abstract for groups lacking medical experiences, such as junior medical students, medical engineers, interdisciplinary researchers, and patients. It is difficult to compare the TMIs with textbook-like anatomical structures.

For example, for medical students, interdisciplinary researchers, medical engineers, and even young physicians, it is difficult to quickly locate and identify some complex lesion areas and subtle anatomical structures in TMIs and establish a direct understanding of the anatomy as senior experts do, especially for low-quality datasets in clinical practice and deformed organs caused by trauma or lesions; for patients and their families, with the increasing awareness of the active participation of the patients and their families in the process of diagnosis and treatment of diseases, it is still necessary for professional doctors to provide guidance or markings to help patients establish a direct understanding of the lesions because ordinary people lack knowledge of anatomy. In addition, the above-mentioned groups are not in the same closed-loop system; when they each obtain a set of TMIs, their intelligent assisted tools should be run in a lightweight, universal, and open scenario, for example, an ordinary personal computer that is readily available and has no special hardware or software environment limitations.

Therefore, we need a technical approach to enable lightweight and rapid registration of TMIs (e.g., CT, MRI) from any patient’s head region in the clinic with the CVH (Chinese Visible Human) [1] dataset and quickly locate the area of interest and present it to “non-professionals” by comparing the high-definition CVH map with the ROI organ location. The crux of the problem is how to perform low-cost, rapid, and lightweight registration of CVH datasets with clinical head TMIs from different modalities, which is unlike the ideal requirements for TMI data quality in laboratory research. In this regard, we propose a simple procedure that can be performed by “non-professionals” to enable fast registration and spatial mapping of TMIs from any patient’s head region in the clinic to the anatomical structure of the CVH.

## 2. Material and Methods

### 2.1. Data Preparation of CVH

The CVH (Chinese Visible Human) dataset is a complete, high-quality digital dataset of the oriental human body developed by the research team of the Department of Digital Medicine at the Army Medical University (Third Military Medical University) [1]. Among them, CVH-2 is the second Chinese Visible Human dataset [2], which has no anatomical structure loss, no obvious pathological changes, is representative of the normal Asian body type and complies with the regulations of the Chinese Ethics Committee.

The resolution ratio of each image of CVH-2 head and neck datum is 3872 × 2048, and the average size of each file is 36 MB [3], with a total of 1018 sequential images, and the total data volume is 1.62 GB after PNG compression. The data spacing of the 1st~800th slices is 0.25 mm, and the data spacing of the 801st~1018th slices is 0.5 mm. Due to the huge amount of data and the data slice spacing being inconsistent, this paper performs spatial size compression and normalization to adapt to the lightweight application on the client side with a final total data amount of 39.8 MB. The detailed pre-processing procedure is shown in Appendix A.

At present, the head and neck region of CVH-2 has been annotated with 128 tissue parts from the parietal bone, frontal bone, dura mater, optic nerve, brainstem, arteries, median ligament of the thyroglossal bone to the esophagus, which can satisfy the universal needs of ROI localization (mapping).

### 2.2. Methods

#### 2.2.1. Overall Process

Because the head and face of the human body will not undergo significant deformation in the short term and are also less affected by physiological movements such as breathing or heartbeat, the registration of the CVH with CT and MRI images in this study can be regarded as a rigid structure registration problem [4,5]. Furthermore, considering that the difference between the human body and air is more obvious in both CT and MRI, this feature enables both CT and MRI to effectively highlight facial regions in surface rendering through appropriate thresholds. Moreover, the face has obvious feature intervals for the registration work [4,5,6,7,8], which is easy for non-professionals to understand. So in this paper, we choose to take the face contour as the basis of the rigid structure. The total registration idea and ROI localization flowchart are shown in Figure 1.

It should be noted that, in order to ensure the authenticity of the clinical image data, the clinical TMI data are usually used as the target data, and the CVH is used as the moving data, which is converged to the TMIs by the CVH.

#### 2.2.2. Exterior Contour Extraction and 3D Reconstruction

3D/3D registration is generally applied between two sets of TMI data or between a set of TMIs and a set of spatial information (e.g., EEG data) [9,10,11,12]. This paper is based on the registration of two sets of continuous tomographic images, TMIs and CVH, which is a typical 3D/3D registration mode. To adapt to the extraction of exterior contours in clinical diversified TMI imaging scanning patterns, it is designed to extract the exterior contour lines by morphological methods (open and close operation, erosion operation, filling, etc.) for each two-dimensional (2D) slice of the original TMIs before 3D reconstruction to remove the impurities. The extraction process is as follows: 1. binarization based on the exterior contour boundaries; 2. filling the closed region; 3. expanding the canvas; 4. filtering the impurities by the opening operation; 5. reducing the canvas; and 6. erosion and subtraction.

The final stable exterior contour extraction effect obtained from TMIs is shown in Figure 2. Among them, Figure 2A shows clinically diverse TMIs, Figure 2B shows the exterior contour extraction effect and C–D shows the reconstruction effect of the exterior contour of B using surface rendering technology. In this paper, the surface contours in Figure 2C,D are used as feature sources for clinical TMIs and CVH data registration.

#### 2.2.3. Introduction to ICP Classic Algorithm

ICP (Iterative Closest Point algorithm) is an algorithm for registering surfaces by iteratively calculating the sum of squares of the residuals of the points corresponding to the surfaces based on the quaternion method [13,14,15]. The basic idea is that there are two different coordinate point sets, P={1,2,…,k} and U={Ui,i=0,1,2,…,n} under the world coordinate system. Set P as the target point set and U as the moving point set and obtain a new point set U’ by continuously rotating and translating the point set U so that the distance between the point set U’ and the homologous point of P is minimized (so that U’ and P overlap as much as possible). U’ can be obtained by the rigid-body geometric transformation Equation (1).
(1)$$\begin{array}{c} U^{’}=RU+T \end{array}$$
where R represents the three-dimensional rotation matrix of the transformed point set U and T represents the translation vector of the transformed point set U’.

The core of this process is to use the minimum root mean square method to calculate the residual sum of squares of the corresponding points between the point sets U’ and P iteratively by continuously correcting R and T to find the minimum error of the root mean square between U’ and P. If the error is smaller than the preset limit value, the iteration is finished, that is, the optimal solution of the registration is obtained.

In the point set transformation, R and T of Equation (1) can also be represented by the chi-square transformation matrix to represent the coordinate transformation relationship. As in Equation (2), we use a 4 × 4 chi-square transformation matrix to spatially transform each eigenpoint (x, y, z) of the point set U, which produces a new point (x’, y’, z’) from which we compute the translation and rotation matrices between these two coordinate systems.
(2)$$\left[\begin{array}{c} x^{’}\\ y^{’}\\ z^{’}\\ 1 \end{array}\right]=\left[\begin{array}{cccc} M_{11} & M_{12} & M_{13} & M_{14}\\ M_{21} & M_{22} & M_{23} & M_{24}\\ M_{31} & M_{32} & M_{33} & M_{34}\\ M_{41} & M_{42} & M_{43} & M_{44} \end{array}\right]\cdot \left[\begin{array}{c} x\\ y\\ z\\ 1 \end{array}\right]$$
where (x,y,z) is the coordinates of a point in the set of moving points, $M_{ij}=\left[\begin{array}{cccc} M_{11} & M_{12} & M_{13} & M_{14}\\ M_{21} & M_{22} & M_{23} & M_{24}\\ M_{31} & M_{32} & M_{33} & M_{34}\\ M_{41} & M_{42} & M_{43} & M_{44} \end{array}\right]$ is the homogeneous transformation matrix, and (x’,y’,z’) is the new coordinate point obtained from the transformation of the moving point set. For the homogeneous transformation of 4 × 4 matrices, the fourth row is [0, 0, 0, 1], used to maintain the homogeneous coordinates of points.

However, due to the complexity and irregularity of the spatial distribution of the feature point sets, the ICP algorithm is easy to fall into the local extreme value problem. For example, the exterior contours extracted from clinical TMIs and CVH are used as two sets of feature point sets, the ICP algorithm is directly used to register from CVH to TMIs, and the registration result is obtained as in Figure 3, where the nose of CVH is registered with the ear of Data 2, completely losing directionality.

This is due to the fact that protruding areas such as the nose and ears, as well as similar gradient areas such as the forehead and occiput, tend to result in false “best” matches during the registration process. This phenomenon is difficult to solve by setting a higher number of iterations because the ICP algorithm stops iterating once it falls into local extremes during the iteration process.

#### 2.2.4. TSPR Method and the Adaptability Improvement of ICP Algorithm

To address this problem, this paper proposes the TSPR method, which solves the local extreme value problem by selecting different feature regions and implementing two or more times ICP registration.

(i)First ICP registration for spatial orientation

Figure 4 [16], so that the target point set P1 and the moving point set U1 converge in direction and obtain the homogeneous transformation matrix $M_{1ij}$ for the motion indication of the moving point set U1.

(ii)Second ICP registration for spatial scaling

After obtaining consistency in the direction of two sets of point sets, select feature points with better spatial correlation between the two sets of data and determine the target point set P2 and the moving point set U2. To prevent the occurrence of local extremum problems, firstly, restore U2 to the motion state of the first ICP registration, and obtain U2’. Then, let U2’ perform a convergence operation towards P2 according to the ICP algorithm to obtain a homogeneous transformation matrix $M_{2ij}$ for the motion indication of the moving point set U2’.

The registration of the spatial orientation is needed only once, while the registration of spatial scale can be progressive multiple times on the basis of the previous registration until a satisfactory spatial registration effect is achieved. Because the purpose of spatial orientation registration is for rough spatial registration between the U2 and the P2 object, once completed, subsequent fine registration (such as deformation registration) usually does not significantly change the overall spatial orientation. Repeating multiple spatial orientation registrations can easily lead to over registration and increase computational costs. And the final homogeneous transformation matrix is the product of the transformation matrices of the previous multiple registration, as shown in Equation (3).
(3)$$M_{ij}=M_{1ij}\;\cdot \prod_{x=1}^{n-1}M_{2ij,\;\;x}$$
where $M_{1ij}$ is the homogeneous matrix obtained after the first registration, $M_{2ij,x}$ is the newly obtained homogeneous matrix after the registration of the moving point set based on the previous transformed orientation, and n represents the number of times that the progressive registration is performed. $M_{ij}$ is the final homogeneous matrix result of the transformed relationship between the moving point set and the target point set, and each voxel point of the CVH dataset will be spatially transformed via $M_{1ij}$ to achieve the final spatial mapping with the clinical image data.

#### 2.2.5. Design of TSPR Interaction

Compared with voxel-based 3D/3D spatial registration, the registration of facial contour in Section 2.2.2 has the advantages of intuitive feature point selection and small data computation. Further combining the theory in Section 2.2.3, the TSPR method can be standardized as “Facial features registration determine spatial direction—facial circular contours registration determine spatial scale”.


**First Step: facial features registration**


The “eye–nose triangle” facial feature rigid transformation is used to determine the spatial orientation and complete the coarse registration. In order to simplify the operation, the vertical mapping of the face is adopted, as shown in Figure 4. At A, the operator selects the corresponding 2D rectangular region on the clinical image and the 3D reconstructed image of the CVH, respectively. At B, the system uniformly collects 28 feature points on the respective surface contour according to a discrete sampling frequency of 4 × 7 and then performs the rigid registration according to the ICP algorithm in C-D to obtain the homogeneous matrix equation $M_{1ij}$, a detailed interpretation of the ICP algorithm can be found in Appendix A.
$$\mathrm{In}\;\mathrm{this}\;\mathrm{example},\;M_{1ij}=\left[\begin{array}{cccc} 0.1533 & -0.9604 & 0.2326 & 207.9560\\ 0.9635 & 0.0930 & -0.2511 & 5.6186\\ 0.2195 & 0.2626 & 0.9396 & -56.8884\\ 0.0000 & 0.0000 & 0.0000 & 1.0000 \end{array}\right]$$

$M_{1ij}$ can directly guide the CVH to perform the spatial transformation to obtain the registration effect of E, where the MRI image is superimposed with the same scene image of the transformed CVH. As seen in Figure 4E, the spatial positions of the two datasets have converged.


**Second Step: facial circular contours registration**


On the basis of the first rigid registration, similarity registration was performed using circular contour features to determine the spatial scale to accommodate the size difference of different people’s heads. Since similarity registration increases the risk of local extremes, the second registration needs to select two sets of circular feature contours with approximately the same data. As shown in Figure 5, the eye–nose triangle is recommended as the circular base point in Figure 5B to obtain the image features that require secondary registration (the circular structure of the face).

The similarity registration is performed at C according to the ICP algorithm to obtain the homogeneous matrix equation $M_{2ij}$. The conversion from Equation (1) to Equation (2) refers to Appendix A.
$$\mathrm{In}\;\mathrm{this}\;\mathrm{example},\;M_{2ij}=\left[\begin{array}{cccc} 0.9059 & -0.0172 & -0.2012 & 30.4818\\ 0.0630 & 0.9026 & 0.2068 & -30.5599\\ 0.1919 & -0.2155 & 0.8821 & 38.3209\\ 0.0000 & 0.0000 & 0.0000 & 1.0000 \end{array}\right]$$

$M_{2ij}$ can guide the CVH to be further transformed on the basis of the previous spatial transformation $M_{1ij}$ to obtain the registration effect of D, where the MRI image is superimposed with the same scene image of the transformed CVH. Figure 5D shows that, compared with the first registration result, the effect after the second registration has been more satisfactory.

According to Equation (3), the homogeneous matrix obtained from the first registration is multiplied by the homogeneous matrix obtained from the second registration, from which the final homogeneous transformation matrix of the whole registration process is obtained as follows:

Final homogeneous transformation matrix result,
$$M_{ij}=\left[\begin{array}{cccc} 0.0782 & -0.9245 & \;0.0259 & \;230.2195\\ 0.9247 & \;0.0777 & -0.0177 & -24.1410\\ 0.0155 & 0.0273 & \;0.9276 & 26.8280\\ 0.0000 & 0.0000 & 0.0000 & 1.0000 \end{array}\right]$$

The final homogeneous transformation matrix$\;M_{ij}$ obtained will be used to guide the subsequent visual representation of the mapping of gross anatomical structures (for details about the CVH anatomical knowledge database, please refer to Appendix A).

The software interaction interface(ver 1.0.0) is shown in Figure 6 and the demonstration video from Appendix A.

Operational Steps Overview:

Step 1: Data Loading

When the user opens this application, the software enters Figure 6A, where the CVH template automatically loads the head contour image of CVH on the left side. After the user selects the data to be projected in Clinical Data (CT/MRI) as prompted, the system will automatically complete the external contour extraction and reconstruct the head contour image.

Step 2: Selection of Facial Feature Points and First Registration

In Figure 6A, the user adjusts the position of the projected contour image using the mouse, then selects the “Select” function key to draw symmetric eye-nose triangle areas on both CVH and Clinical Data. The system will automatically complete the collection of feature points (see blue and red points in Figure 6A). Then, as prompted, click the “First registration” function key in Figure 6A. The “Registration situation” area on the right will display the spatial relationship after the first registration (blue points represent the original facial feature positions of CVH, red points represent the original facial feature positions of Clinical Data, and green points show the spatial relationship of blue points after registration).

Step 3: Selection of Circular Contour Area and Second Registration

In Figure 6B, the user uses the trackbar to select the circular contour areas for both CVH and Clinical Data. Then, as prompted, click the “Second registration” function key in Figure 6B. The “Registration situation” area on the right will display the result of the complete 3D image of CVH converging towards the spatial position of the complete 3D image of Clinical Data after the second registration.

Step 4: Spatial Mapping

Click the “Mapping” button on the right side of Figure 6B to pop up dialog boxes as shown in Figure 6C,D. Users can input organ names in the ROI mapping text box and check them in the organ list for quick mapping in the image area (as shown by the red areas in Figure 6C,D. Users can adjust the image perspective in 3D view using the mouse in the image areas of Figure 6C,D, or view the transverse, coronal, and sagittal planes separately through the trackbar in the upper right corner.

For more operational details, please refer to the video from Appendix A.

## 3. Results

### 3.1. Selection of Test Data

In this paper, six sets of real, representative clinical head serial TMI data were selected from the PACS system of Southwest Hospital of AMU to validate the effectiveness of the methodology. To protect patient privacy, this article erases the label information of DICOM images, and their main parameters and characteristics are shown in Table 1.

Table 1 shows that Data 1, 2, 3, 4, 5, and 6 have different head positioning, scanning intervals, scanning range, and slice quantity, have different ROI organs with their respective diagnostic reports, and come from five patients with different face shapes, so the applicability of the method of this paper can be evaluated to a certain extent, and each group of data was processed three times by different people so that a more objective evaluation of reproducibility can be obtained.

### 3.2. Registration Effect of Clinical Data and CVH

Currently, there is no unified evaluation scale or gold standard to evaluate the registration between heterogeneous multimodal images [17]. We performed a 3D reconstruction of the two datasets in the same scene to evaluate the effectiveness of registration based on the fit between the two datasets. Since CVH is a 24-bit true-color bitmap dataset, which is completely different from the imaging features of CT and MRI, we designed the two volume rendering schemes of coloring ranges and transparency to clearly separate them.

The results of the same scene after Data 1, 2, 3, 4, 5, and 6 were registered using the methods of this paper are shown in Figure 7. The left column in Figure 7 represents the original positioning, while the two columns on the right show different perspectives of the results after registration using the method proposed in this paper. It can be seen that from Data 1 to Data 6, our method can successfully overcome the influence of local extreme values. Among them, the clinical data selected in Data 5, unfortunately, has missing teeth, and Data 6 has a large area of bone defect near the location of the left temporalis muscle, but our method still performs good registration on the areas above the “eye–nose triangle”. If we focus our observation on the tooth area, we can choose the “oral–nasal triangle area” instead of the “eye–nose triangle area” so that we can obtain a registration result that pays more attention to the “oral–nasal triangle area”. In a word, Through different perspectives, it is subjectively believed that the registration results reach the basis of the final ROI targeting aim of this paper.

### 3.3. Five-Point Scale Method Based on the Mapping Effect of Typical Anatomical Structures

The anatomical structures of the optic nerve, pituitary gland, cerebellum, brainstem, and temporalis muscle were selected as ROI organs for projection with respect to the location of the lesion seen on imaging for each of the six datasets, and representative results are shown in the first and second columns of Figure 8.

In Figure 8, different colors in the first and second columns are used to represent the positions of different ROI regions in Data 1–6; all six sets of data have lesion areas related to ROI, which poses a challenge to traditional image registration methods. However, as shown in Figure 8, the method proposed in this paper can effectively adapt to these changes, demonstrating enhanced adaptability in processing image data affected by lesions.

Furthermore, we invited one adult with a non-medical background (role of patients) and two third-year biomedical engineering students (role of medical students and engineers) to use their respective computers to perform a total of 18 ROI localization operations on the six sets of data to obtain the localization of the organs as shown in Figure 8, and then a radiologist was invited to confirm the localization position. Finally, a five-point scale for subjective evaluation results from the 18 operations was designed and collected. See Table 2.

The application of the five point scale provides key insights into the usage experience of participants. For the accuracy of interaction behavior, Table 2 shows that out of a total of 18 operations, 15 operations are smooth operations with one-time success; the other three operations were due to the initial impression for the first time operating without training, but at least all achieved the mapping goal. For the efficiency of interaction behavior, there are 17 operations that can be completed in 1 min; only one operation was completed in 2 min because the operator wanted to perform more observations the first time using it. For the matching degree of organ mapping, the total number of operations that perfectly match Data 1 to 6 is 15; the rest are option B (mostly match), which can meet the universal needs of medical students, medical engineers, interdisciplinary researchers, and other groups. The results show that the TSPR method proposed in this paper has accurate registration results, is simple to operate, takes no more than 1 min on average for each operation, and has good adaptability and repeatability to clinical imaging data, which can satisfy the research purpose of this paper.

### 3.4. The Quantitative Analysis Method Based on ROI Area Coverage, the Mapping Effect of Typical Anatomical Structures

For the six sets of data in Figure 8, we invited a doctor with 12 years of work experience from the Radiology Department of Southwest Hospital to annotate the optic nerve of Data 1 (CT) and Data 2 (MRI), pituitary gland of Data 3, the cerebellum of Data 4, and brainstem of Data 5, and temporalis muscle of Data 6 slice by slice according to the cross-section. The annotation effect is shown in the third column of Figure 8. At the same time, another doctor with 15 years of work experience was invited to confirm and ensure the accuracy of the ROI area division and further design Equation (4) for quantitative calculation.
(4)$${Recall}_{i}=\frac{{ROI}_{c,i}\cap {ROI}_{m,i}}{{ROI}_{m,i}}$$

Among them,

${ROI}_{c,i}$ refers to the area of the ROI region automatically mapped by the system on the i-th slice;

${ROI}_{m,i}$ refers to the area of the ROI region marked by the doctor on the i-th slice.

${Recall}_{i}$ represents the proportion of the intersection between the ROI area mapped by the system on the i-th slice and the ROI area annotated by the doctor in the ROI area annotated by the doctor. The higher the recall value, the better the directionality.

We use the ROI data annotated by doctors as a benchmark, select the range of layers in the tomographic images with ROI, and automatically calibrate the ROI area by comparing and extracting the corresponding layer of the first operation data of adult subjects obtained in Section 3.3, and jointly input it into Equation (4) for calculation to obtain the results shown in Table 3.

In Table 3, Data 1 to Data 6 cover data of different modalities, such as incomplete-scan, thin-layer CT or non-thin-layer CT, MRI, etc., for different individuals, and the organ ROI has significant differences in body shape (detailed in Table 1). Therefore, the coverage layer range of ROI varies greatly, but the recall values are very satisfactory.

According to the data in Table 3, the maximum, minimum, median, and average values of the recall with prominent correlation for each group of data are obtained, as shown in Table 4, and further calculate the 95% confidence interval using Equation (5).
(5)$${CI}_{95\%}=\bar{X}\pm 1.96\times \frac{s}{\sqrt{n}}$$

Among them,

$\bar{X}$ refers to the average recall rate;

s refers to the standard deviation of the recall rate;

n refers to the sample size.

From Table 4, it can be seen that the TSPR fast localization proposed in this article can always cover the ROI areas that need to be indicated and achieve the goal of fast localization.

## 4. Discussion

### 4.1. Adaptability Requirements for Test Data

The multimodal tomographic images in this paper mainly include clinical TMIs (CT and MRI) and CVH datasets.

The ROI of Data 1 and 2 showed an oval-shaped nodule lesion on the inner side of the left optic nerve behind the orbital ball, both of which were radiological findings. However, there are essential differences in the imaging modality between the two sets of data: Data 1 is a CT dataset, which clearly displays rigid structures such as bones; Data 2, on the other hand, was obtained through MRI scanning, which is a better display of soft tissues. Although everyone is concerned about the optic nerve and adjacent oval-shaped nodule lesion, the performance of the two modal data poses challenges in universal image recognition. In addition, Data 1 has a higher number of slices (121 slices) and more detailed slice intervals (1 mm), while Data 2 has the opposite, with only 18 slices and a 3 mm interval. In order to evaluate the robustness of the proposed method in processing non-thin layer scanning and low-resolution image data, the two challenging image data mentioned above were selected as validation objects.

The ROI of Data 3 was clinically diagnosed as a nodular lesion on the left side of the sellar region, characterized by distinct nodular images in the imaging data. And the ROI of Data 4 was clinically diagnosed as structural abnormalities and disorders in the cerebellar region. Due to the potential impact of atypical structures on conventional image registration methods, the analysis of these two datasets is particularly important for verifying whether the proposed method can maintain the correct ROI mapping ability in different clinical disease scenarios. This comparative analysis will help demonstrate the robustness and effectiveness of our method in handling TMI data with deformities or abnormal structures. Ensuring the accuracy of registration methods under various pathological conditions is crucial for further analysis and diagnosis.

In addition, Data 5 contains a set of high-quality clinical CT images, but it must be pointed out that in actual clinical environments, the majority of imaging data often fail to meet such high-quality standards in the PACS system. It means that CT or MRI with limited scanning area, 3–5 mm thick tomographic images actually stored in PACS, ROI localization target for deformed lesions, is the common form of clinical TMI data.

Data 6 shows significant external deformation, as the patient has suffered from brain trauma. Imaging shows large areas of bone loss in the left temporal parietal bone adjacent to the left temporal muscle, edema and softening of the left temporal parietal lobe, and slight swelling of the temporal muscle. After 3D reconstruction, there is a significant protrusion on the left external part of the brain, increasing the diversity of localization targets.

The selection of the above data could enable this study to evaluate the performance stability of the proposed method when facing clinical imaging materials below the ideal quality of laboratory research. By verifying the processing ability of this method for images of different quality, we can further understand its potential application and adaptability in practical clinical scenarios.

### 4.2. Discussion on the Generalization Performance of Diversified Images with Significant Pathological Changes

The focus of this article is on the rapid localization of organs or lesions within the head. From the experimental results, it can be seen that using a two-stage registration method of facial features and circular features can effectively capture the area of concern for the operator. But, testing their method on a diverse set of images, especially those with significant pathological changes, is the focus of verifying clinical robustness. Therefore, in the selection of cases from Data 1 to Data 6, we strive to find representativeness in terms of the diversity of imaging modalities, the diversity of imaging quality, and the impact of lesion changes on ROI interval occupancy (detailed in Section 4.1 and Table 1).

From the 58th slice of the radiologist’s annotation in Data 1 (CT) of Figure 8, it can be seen that compared to the right optic nerve (marked in red), the left optic nerve (marked in green) is severely deformed due to tumor compression. From the TSPR mapping of optical nerves in Figure 8, the right optic nerve (red projection) covers the optic nerve area very well, almost consistent with the doctor’s standard, while the left optic nerve (blue projection), although deviating, points out the correct orientation and lesion of the optic nerve, which is beneficial for non-professionals to compare and observe, and achieve the goal of rapid positioning for navigation.

As the same patient, Data 2 (MRI) also achieved navigation effects consistent with Data 1.

There is a nodular shadow on the left side of the saddle area in the ROI region of Data 3, considering the possibility of pituitary adenoma. Data 4 has postoperative cervical changes and abnormal structural disturbances in the cervical region, and Data 5 has cerebral softening foci in the left part of the brain, demyelinating changes in the cerebral white matter. Therefore, the above cases all have a certain degree of deformation, and even Data 5 shows complete tooth loss and external facial features. However, the experimental results all showed the ability to quickly locate issues.

Furthermore, we introduced Data 6, which involves a patient with brain trauma. Taking the 10th slice of the original image in Data 6 of Figure 8 as an example, the radiological findings show a large bone defect in the left temporoparietal bone adjacent to the left temporalis muscle, left temporoparietal lobe cerebral edema, formation of a softening focus, and slight swelling of the temporalis muscle. From the 10th slice of the radiologist’s annotation in Data 6 of Figure 8, it can be seen that compared to the right temporalis muscle (marked in red) and the left temporalis muscle (marked in green) is significantly deformed due to trauma. After 3D reconstruction, there is a noticeable bulge on the external left side of the brain. After completing the registration and positioning of the left and right temporalis muscles using the method proposed in this paper, from the TSPR mapping of the temporalis muscle in Figure 8, both the right (yellow projection) and left (red projection) temporalis muscle covers the temporalis muscle area very well, almost completely covering the area outlined by the doctor. The area adjacent to the left temporalis muscle shows significant deformation, which is beneficial for non-professionals to compare and observe and achieve the goal of rapid positioning for navigation. The recall value for positioning in this slice reached 83.53%. According to the statistics in Table 3 and Table 4, the positioning of other slices is also good.

The above case, as evidenced by the results shown in Figure 8 and the statistical data presented in Table 2 and Table 4, demonstrates that this method possesses strong robustness and versatility.

### 4.3. Advantages of the Method Proposed in This Study

(i)Two-step strategy for image registration

In the field of medical image registration, a two-step registration process (including coarse registration and fine registration) is a commonly used and effective strategy. This method can balance the accuracy and computational efficiency of registration. This strategy is widely used in medical image registration; for example, Liu et al. [18] used Mimics 20.0 software to perform two-step registration of CBCT dental arch and optical scanning dental models in non-bite and bite states using global registration and local registration, respectively. Chen et al. [19] proposed an improved registration method based on the internationally renowned brain functional imaging software package SPM 12, which also follows the concept of course to fine registration and effectively improves the accuracy of registration. Therefore, we can see that in the field of medical image registration, the two-step registration process of using one coarse registration followed by one fine registration has become a widely recognized theoretical framework. This method not only balances the accuracy and computational efficiency of registration but also effectively handles complex medical image registration problems. The TSPR method proposed in this article is based on this theoretical framework, which achieves coarse registration through facial feature registration and fine registration through facial circular contour registration, thus achieving efficient and accurate registration results. Therefore, the TSPR method proposed in this article also adopts a similar two-step strategy but focuses more on the registration of different human heads with the CVH template and provides a templated registration scheme: the first step is to determine the spatial direction through coarse registration; the second step is to achieve precise registration and determine the spatial scale.

(ii)Selection of feature points

In the selection of feature points, they are usually divided into anatomical landmarks and artificial landmarks [20]. Anatomical landmarks refer to naturally existing reference points in the human anatomical structure. These points are typically based on easily identifiable and locatable features in biological anatomy, such as eye corners, the tip of the nose, earlobes, etc. These landmarks are commonly used as reference points in medical imaging, anatomical studies, and surgical navigation because they have relative consistency and reproducibility across populations. Artificial landmarks refer to manually defined and set reference points. These points may not naturally exist but are added to the body or images through certain methods or devices (such as markers, sensors, etc.) to assist in localization or registration in medical imaging. The use of these landmarks may introduce some bias due to their subjective nature. However, in situations requiring high-precision localization, they can provide valuable information. Due to the subjectivity of artificial landmarks, natural anatomical landmarks such as the inner and outer corners of the eyes and nose have become commonly used features in neurosurgery [4,20,21]. The “eye–nose triangle” we chose is based on these stable facial features. These features are easy to identify and provide a stable registration basis. On the basis of facial feature registration, we introduce circular contour registration to solve the problems that may arise from using only facial features. Lindseth et al.’s [20] study suggests that using only facial and ear features may affect the registration accuracy of other parts of the head. By adding circular contour registration, we have expanded the registration range and improved the accuracy of overall head registration. We recognize that the accuracy of using anatomical landmarks or surface registration alone may be lower than registration based on artificial markers. Therefore, our TSPR method combines anatomical landmarks (eye–nose triangle) and surface registration (circular contour), aiming to improve the accuracy and robustness of registration.

(iii)Comparison of Registration Methods

Although the current popular machine learning-based organ localization has the advantage of high recognition accuracy, it is difficult to have a complete model to cover the complex and diverse or pathological organs in the human body part, such as meeting the localization needs from Data 1 to Data 6. Moreover, this large model training approach is quite demanding on data modality, data quality, and application scenarios [22,23,24,25,26,27,28], and has some stability and interpretability issues when facing complex clinical data environments. For example, the incomplete head data in Data 1 (the radiologist only scanned some areas because the actual ophthalmic examination only focused on the eye region) and the non-thin-layer data in Data 4 (which is stored on-demand under the limited storage space of PACS), all put high challenges on the robustness and generalization ability of the model.

In terms of registration, machine learning based registration methods also have similar issues with organ localization. By contrast, the marking points-based registration methods have greater practicality in clinical practice. Literature mainly focuses on studying the local features auxiliary extraction of a certain anatomical region [29] or interior features manual marking of organs by radiologists with the help of professional software [30,31,32]. The former improves performance rapidly with the help of machine learning but struggles to adapt to the diverse clinical data modality and data content; the latter requires users to have specialized anatomical knowledge, clinical experience, and software training, which makes it difficult to support the lightweight application scenario of this paper.

As a specialized area of machine learning, deep learning technology focuses on neural network models, making it suitable for solving more complex tasks and problems. Zhang et al. [33] proposed a two-step registration method based on deep convolutional neural networks for the registration of multimodal retinal images. This method also includes two steps: coarse registration and fine registration. However, our method is based on traditional surface anatomical features for registration, while Zhang et al.’s method [33] relies entirely on deep learning techniques and solves the problems of inconsistent multimodal images and lack of labeled data through an unsupervised learning framework. Yang et al. [34] proposed a deep learning brain image registration method based on open–close operation morphology, which has achieved significant results in improving the accuracy of brain MRI registration, especially in complex boundary areas and small, narrow brain regions. Although these methods perform well on specific tasks, they still face some challenges. These challenges include the need for large datasets, which may not always be easily achievable in real clinical environments; it is difficult to apply the demand for computing resources to lightweight application scenarios; the generalization performance is limited, and in fact, these literature are only applicable to specific tasks. Zhang et al.’s work [33] only has good performance in the field of retinal images, but the cross-domain generalization ability is limited. Yang et al.’s work [34] achieved good results on public datasets, but it cannot be applied to registration between different individuals, and deep learning models struggle to achieve lightweighting. Although Zhao et al. [35] proposed a medical image registration model called PPCTNet based on CNN and Transformer in parallel, which reduced the model parameters by 22 M and computational complexity by 500 G in MRI registration of the same brain, striving to improve lightweight performance, the issue of lightweight models remains a long-term research process when facing brains of different individuals.

In comparison, the TSPR method adopts a two-step registration method in terms of strategy and combines anatomical landmarks of external contours and artificial markers in feature point selection. It is not only suitable for multimodal images (such as CT, MRI, and CVH datasets) and different individuals but also for handling incomplete scans and non-thin-layer data (such as Data 1 and Data 4). Meanwhile, TSPR has a spatial reduction. Occupying less than 10 M and with an average user operation time of less than 1 min, both in terms of space occupation and computational complexity, it is far smaller than deep learning methods. In addition, the TSPR method is easy to operate, with an average operation time of no more than 1 min, and has good flexibility to adapt to ROI tissue/organ deformation caused by trauma, lesions, etc. Compared to “black box” deep learning models, the working principle of the TSPR method is clearer and easier to understand and accept. These characteristics make the TSPR method significantly advantageous in practical clinical applications. The results of Table 2 and Table 4 show that the TSPR interaction method based on the rigid structure of exterior contours proposed in this paper has the characteristics of simple and intuitive operation, no professional requirements for the user, no need for training, and better meets the low-cost and universal requirements of junior medical students, medical engineers, interdisciplinary researchers, and general patients.

Meanwhile, facing the deformation of ROI tissues/organs due to trauma, lesions, etc., in real clinical data, the method in this paper has stronger data adaptability through feature registration of the overall external contour with templatized head registration and ROI organ mapping. Moreover, compared to traditional ICP algorithms, the TSPR method proposed in this paper demonstrates significant advantages in avoiding getting stuck in local extremum problems. For the other parts of the body, our method provides users with a more flexible interaction space for feature selection. For example, when the user is interested in the oral cavity and related parts of Data 5, the user can re-register the features according to the oral-nasal triangle. In fact, it has become an effective auxiliary tool for Junior medical students and interdisciplinary researchers in our college.

Of course, there are still some areas that can be improved. For example, this work mainly focuses on clinical TMIs, which are composed of cross-sectional data, but there are a large number of sagittal or coronal scanning results in MRI scans. To incomplete or non-thin-layer sagittal/coronal scan data in clinical environments, the applicability of this method is planned to be studied in the next work. In addition, the interaction of TSPR currently requires manual operation, and although the operation steps are simple. But for some patterned operations, facial recognition technology can be studied to automatically extract facial features and facial exterior features from the 3D facial reconstruction results of TMI without manually selecting feature regions, achieving complete automated interaction and further simplifying the operation process.

Although the TSPR method demonstrates excellent performance and adaptability in processing various clinical data, its true value still needs to be validated in actual clinical environments. Therefore, it is necessary for us to further explore the clinical application prospects of this system to fully understand its potential impact and practical value.

### 4.4. Clinical Application Prospects of the Software

Through experiments and applications in the digital anatomy classroom at the Army Medical University, the rapid head organ localization system developed in this study shows potential for clinical application. This software is based on the TSPR method proposed in this article and has the following characteristics closely integrated with the core contributions of the TSPR method:(i)Applicability and Data Processing Capability:

As described in Section 4.3, the TSPR method is capable of handling incomplete scans (such as Data 1) and non-thin-layer data (such as Data 4). This feature is particularly important in clinical environments as it enables the system to adapt to various practical medical scenarios, including images of different modalities such as CT and MRI.

(ii)Adaptability:

Unlike most studies that focus on segmentation for specific organs or diseases, our work has better environmental adaptability and resilience. For instance, in the presence of shape-abnormal lesions, it can swiftly locate the target ROI organs (as seen in Data 1, 2, and 6). This includes accurately identifying the normal right optic nerve and rapidly pinpointing the left optic nerve compressed by a tumor. It also includes the ability to locate injury areas via the temporal muscle in cases of external trauma.

(iii)Usability and operational efficiency:

The design of the TSPR method simplifies the operation steps and makes the system interface intuitive and easy to use. This enables non-professionals to quickly and effectively perform registration tasks without the need for extensive training or specialized anatomical knowledge. This characteristic is consistent with the fast and lightweight goals pursued by the TSPR method.

(iv)Potential educational value and interdisciplinary applications:

Although further research and clinical validation are needed, this system can provide valuable support for medical students and non-professionals (such as patients, medical engineers, and interdisciplinary researchers) to more effectively interpret and utilize medical imaging data. By comparing the patient’s TMI data with high-definition CVH maps, users can quickly locate areas of interest, which is consistent with the original intention of the TSPR method to provide a more intuitive understanding.

(v)Potential of Digital Public Health and Medical Education:

The rapid processing capability of the TSPR method (as mentioned in Section 3.3, with an average operating time of no more than 1 min) has demonstrated enormous potential for the system in the fields of digital public health and medical education. It provides patients with better channels for informed services while also offering new possibilities for computer-assisted education.

## 5. Conclusions

In this paper, from the perspective of protecting patients’ right to know and facilitating the browsing of clinical TMIs by junior medical students, engineers, and interdisciplinary researchers, a universal TSPR method is proposed. This method enables the tomographic data (e.g., CT, MRI) from the head region of any patient in the clinic to be rapidly regionally registered and spatially mapped with the CVH so that the abstract clinical TMIs can be compared with the high-definition maps of the CVH, and to achieve the purpose of rapidly locating the organs in the ROI. Compared to most studies that have requirements on the quality of image data in laboratory environments, this TSPR method can adapt to incomplete and non-thin-layer quality in real clinical data (such as Data 1–6 in the article, including CT or MRI with limited scanning area, 3–5 mm thick tomographic images actually stored in PACS). Moreover, Data 1–5 represent most disease scenarios for observing internal organs or lesions in the head and can also adapt to the deformation problem of ROI caused by trauma and lesions to a certain extent, and Data 6 represents the localization requirements when head trauma has a certain impact on the external contour structure. The experimental results confirmed its practicality and effectiveness from both subjective and objective perspectives. Also, if there are significant facial deformities, the TSPR method proposed in this paper allows users to avoid severely deformed positions between TMIs and CVH. Users can freely choose feature points for determining spatial orientation and circular structures for determining spatial scale. In summary, this method can run on ordinary personal computers without any special software and hardware configuration in a low-cost way, enabling efficient image reading, computer-aided education, and well-informed rights of patients, highlighting the potential of digital public health and medical education.

## Figures and Tables

**Figure 1 bioengineering-11-00891-f001:**
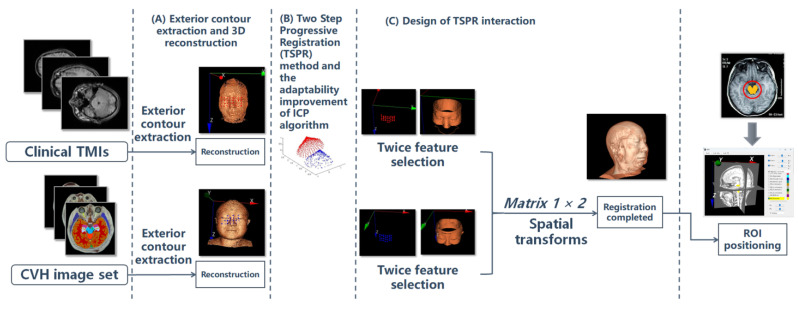
Overall flowchart based on head and face registration and ROI localization. Figure 1 depicts the overall computing logic. It mainly consists of three steps: (**A**) exterior contour extraction and three-dimensional (3D) reconstruction; (**B**) the proposal of the TSPR and the adaptation enhancement of the ICP algorithm; (**C**) the design of the TSPR interaction; and finally, the completion of the spatial transformation of the volume data and the presentation of the ROI according to the registration results. Among them, the registration technology itself adopts the classic ICP algorithm, but the external contour extraction, 3D reconstruction, TSPR method, and adaptive improvement of the ICP algorithm are all original methods independently developed by us.

**Figure 2 bioengineering-11-00891-f002:**
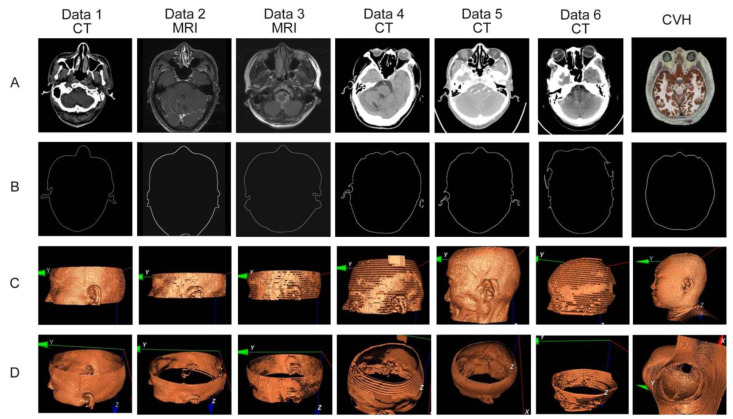
Exterior contour extraction and visualization reconstruction of four representative clinical head CT scan sites (Line (**A**) shows different TMIs used in clinical applications, while line (**B**) displays the extraction results of exterior contours. The surface rendering technique was used to reconstruct the external contour in lines (**C**) and (**D**) using line (**B**), and used as the feature source for the registration of clinical TMI and CVH data in this paper).

**Figure 3 bioengineering-11-00891-f003:**
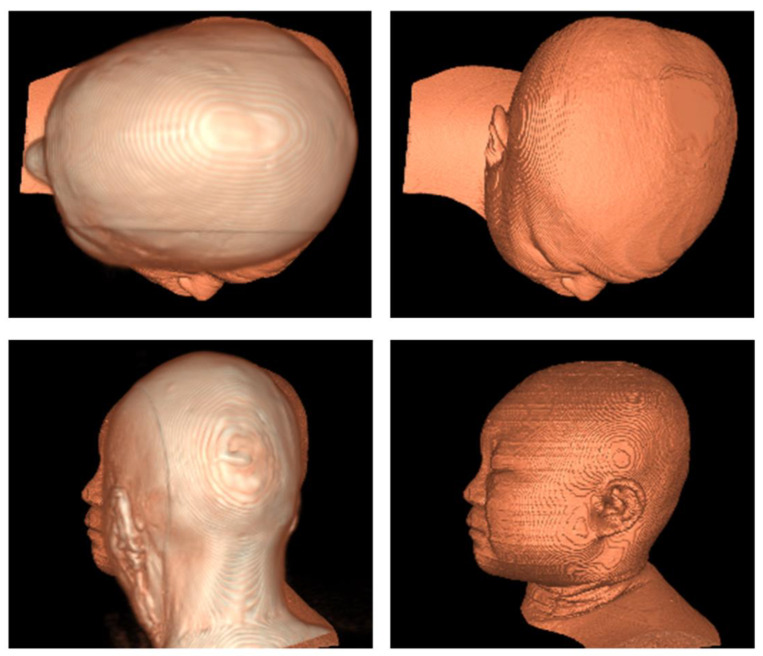
Local extreme diagram (on the left is the 3D reconstruction of the same scene between the registered CVH and TMIs; on the right is the position and morphology of only CVH from the same perspective).

**Figure 4 bioengineering-11-00891-f004:**
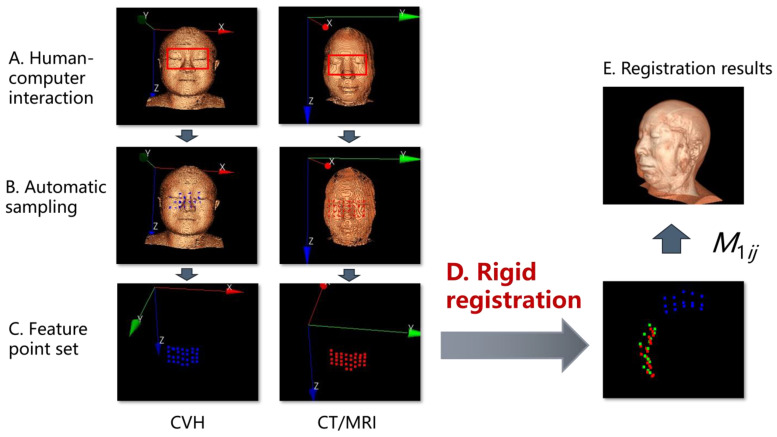
First registration: Facial features for spatial orientation.

**Figure 5 bioengineering-11-00891-f005:**
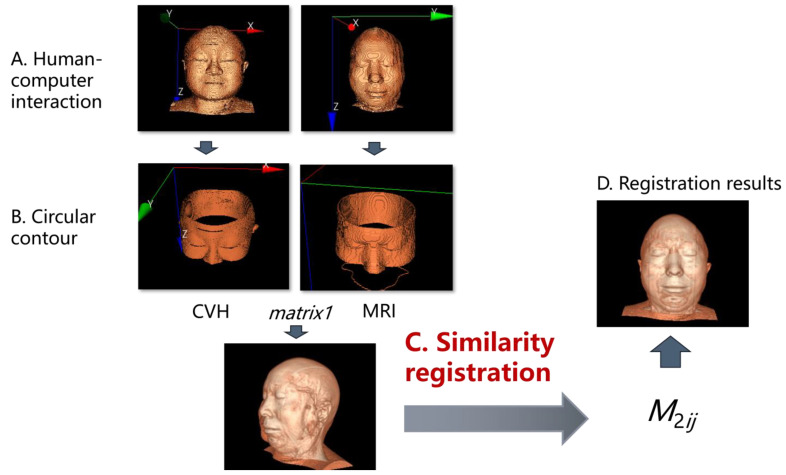
Second registration: facial circular contour for spatial scale.

**Figure 6 bioengineering-11-00891-f006:**
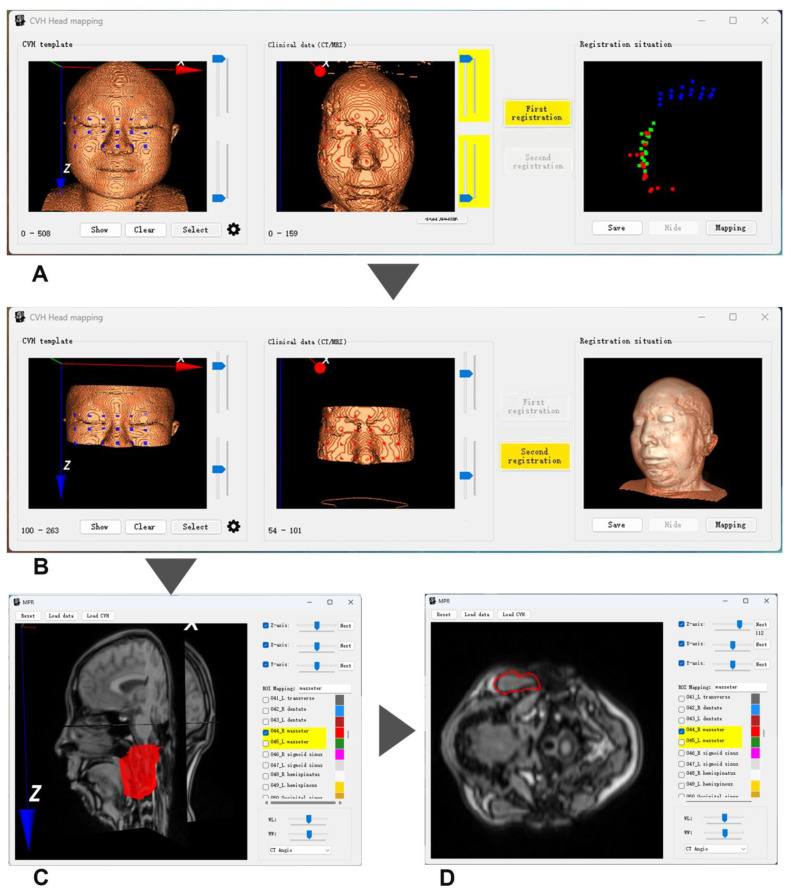
Software interaction interface (Overview of operating steps: (**A**) Data loading and exterior contour reconstruction; (**B**). Facial feature point selection and first registration; (**C**). Selection and secondary registration of circular contour areas; (**D**). Spatial mapping and perspective adjustment. the red area in C represents the right masseter muscle).

**Figure 7 bioengineering-11-00891-f007:**
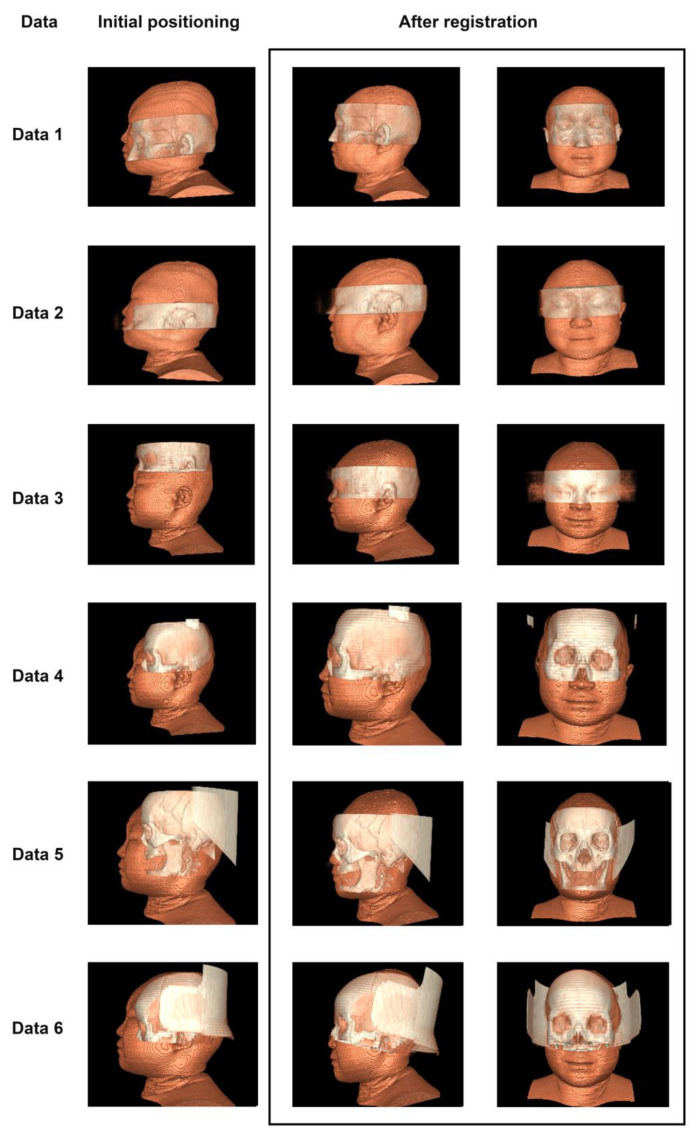
The registration effect between clinical TMIs and CVH, with light colors representing clinical TMIs.

**Figure 8 bioengineering-11-00891-f008:**
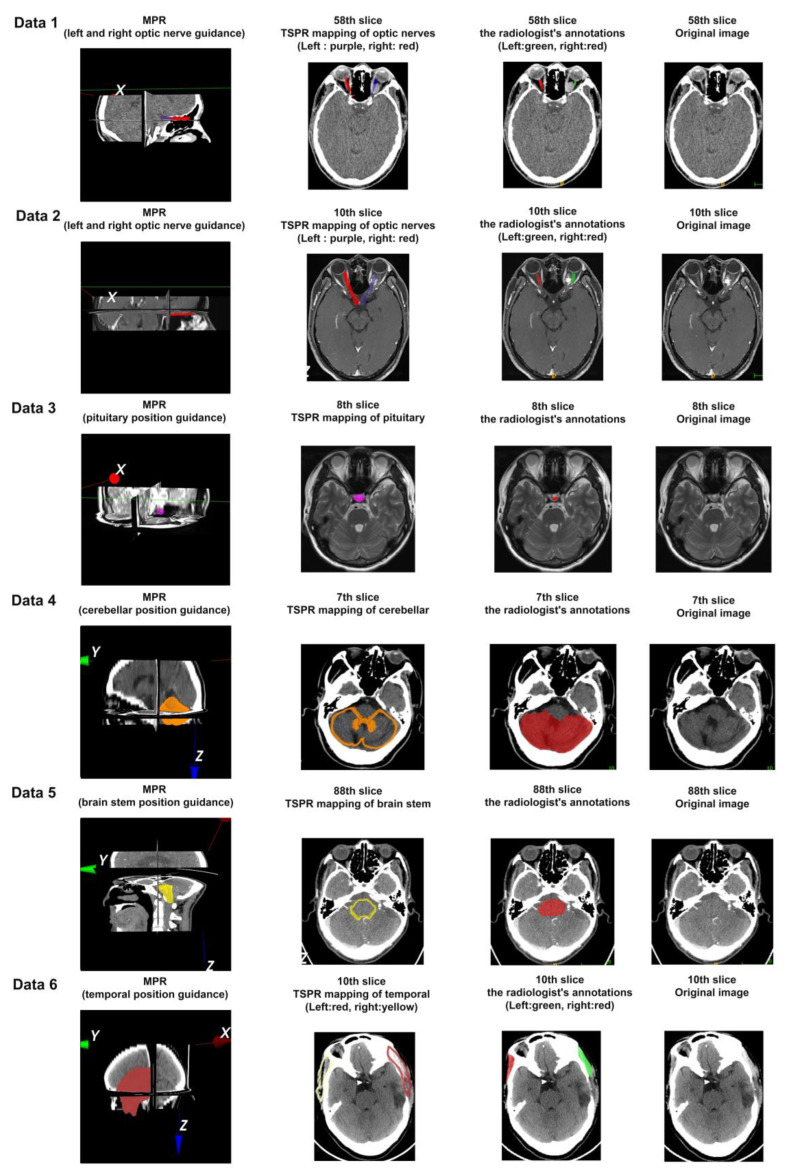
Mapping results of ROI structure between CVH and five sets of TMIs. (The annotated colors are used to indicate the ROI areas).

**Table 1 bioengineering-11-00891-t001:** Parameters of the test dataset.

	Modality	Parameter	MPR of Scan Range
**Data 1**	**CT**	Slice Spatial Resolution: 512 × 512 Slice Quantity: 121 Pixel Spacing: 0.4336 mm\0.4336 mm Thickness: 1.0 mm ROI from radiological description: An oval-shaped nodular shadow can be seen on the inner side of the left ***optic nerve*** behind the orbital ball, suspected to be a hemangioma	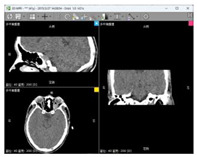
**Data 2**	**MRI**	Slice Spatial Resolution: 288 × 384 Slice Quantity: 18 Pixel Spacing: 0.625 mm\0.625 mm Thickness: 3.0 mm ROI from radiological description: An oval-shaped nodular shadow can be seen on the inner side of the left ***optic nerve*** behind the orbital ball, suspected to be a hemangioma	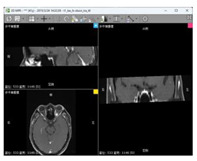
**Data 3**	**MRI**	Slice Spatial Resolution: 512 × 512 Slice Quantity: 15 Pixel Spacing: 0.4296875 mm\0.4296875 mm Thickness: 3.5 mm ROI from radiological description: Nodular shadow on the left side of the saddle area, considering the possibility of ***pituitary adenoma***	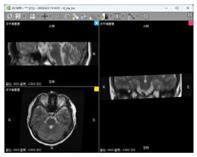
**Data 4**	**CT**	Slice Spatial Resolution: 512 × 512 Slice Quantity: 25 Pixel Spacing: 0.430 mm\0.430 mm Thickness: 5 mm ROI from radiological description: postoperative cerebellar changes, abnormal structural disturbances in the ***cerebellar region***, occipital bone showing postoperative changes	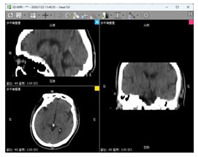
**Data 5**	**CT**	Slice Spatial Resolution: 512 × 512 Slice Quantity: 177 Pixel Spacing: 0.401 mm\0.401 mm Thickness: 1 mm ROI from radiological description: cerebral softening foci in the left part of the ***brainstem***, demyelinating changes in the cerebral white matter	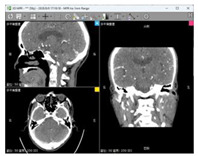
**Data 6**	**CT**	Slice Spatial Resolution: 512 × 512 Slice Quantity: 30 Pixel Spacing: 0.46289 mm\0.46289 mm Thickness: 5 mm ROI from radiological description: large area of bone defect in the left temporoparietal bone adjacent to the left ***temporalis muscle***, edema in the left temporoparietal lobe of the brain, formation of softening lesions, and slight swelling of the ***temporalis muscle***.	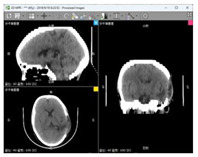

**Table 2 bioengineering-11-00891-t002:** Statistical results of subjective evaluation scale based on CVH gross anatomical structure mapping.

Theme	Five-Point Scale Design				
	A	B	C	D	E
Accuracy of TSPR interaction behavior (**A** smooth operation, one-time success, **B** relatively smooth operation, some steps need to be repeated, **C** achieve mapping objectives, **D** not very useful, **E** ineffective)	15	2	1		
Efficiency of TSPR interaction behavior (**A** can be completed in 1 min, **B** can be completed in 2 min, **C** can be completed in 3 min, **D** is cumbersome, **E** cannot complete the operation)	17	1			
Degree of match between the mapped region of the ROI of the CVH and datasets (**A** perfect match, **B** mostly match, **C** half match, **D** less than half, **E** not valid at all)	15	3			

**Table 3 bioengineering-11-00891-t003:** Statistical results of Recall calculation based on CVH gross anatomical structure mapping.

Data 1	Layer	57	58	59	60	61	62	63	64	65
	Recall	0.8894	0.8366	0.8277	0.8601	**0.8302**	0.8637	0.8949	0.9350	0.9384
Data 2	Layer	10								
	Recall	0.8838								
Data 3	Layer	8								
	Recall	1.0								
Data 4	Layer	2	3	4	5	6	7	8	9	10
	Recall	0.7320	0.8894	0.9369	0.9533	0.9640	0.9115	0.8795	0.9447	0.9503
	Layer	11	12							
	Recall	0.9711	0.4417							
Data 5	Layer	67	68	69	70	71	72	73	74	75
	Recall	1.0	1.0	1.0	1.0	1.0	0.9965	0.9203	0.9937	0.9976
	Layer	76	77	78	79	80	81	82	83	84
	Recall	0.9879	0.8713	0.8813	0.9934	0.9882	1.0	1.0	1.0	0.9784
	Layer	85	86	87	88	89	90	91	92	93
	Recall	0.9876	0.9485	0.9229	0.9497	0.8786	0.8096	0.7717	0.5987	0.6247
	Layer	94	95	96	97	98	99	100	101	102
	Recall	0.6731	0.5881	0.5374	0.6125	0.6908	0.7283	0.6901	0.6126	0.6586
	Layer	103	104	105	106	107	108	109	110	111
	Recall	0.6325	0.6547	0.5833	0.5724	0.6558	0.7003	0.7737	0.7672	0.7104
	Layer	112	113	114	115	116	117	118		
	Recall	0.6720	0.7466	0.7578	0.7619	0.7796	0.8255	0.6970		
**Data 6**	Layer	2	3	4	5	6	7	8	9	10
	Recall	0.8026	0.7408	0.6842	0.6884	0.7196	0.7558	0.7716	0.8242	0.8353
	Layer	11	12	13	14	15	16	17	18	
	Recall	0.8367	0.7917	0.7782	0.7313	0.7641	0.7797	0.7882	0.8368	

**Table 4 bioengineering-11-00891-t004:** Distribution of Recall values for each set of data obtained from Table 3.

	Maximum Recall Rate	Minimum Recall Rate	Median Recall Rate	Average Recall Rate	95% Confidence Interval
Data 1	93.84%	82.77%	86.37%	87.46%	[84.716%, 90.204%]
Data 2	88.38%	88.38%	88.38%	88.38%	
Data 3	100%	100%	100%	100%	
Data 4	97.11%	44.17%	93.69%	87.04%	[78.392%, 95.688%]
Data 5	100%	53.74%	90.08%	82.79%	[75.234%, 90.346%]
Data 6	83.68%	68.42%	77.97%	78.88%	[75.418%, 78.942%]

## Data Availability

If there are reasonable requests, data can be requested from the corresponding author.

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
